# Comparison of Typical Controllers for Direct Yaw Moment Control Applied on an Electric Race Car

**Andoni Medina \*** , **Guillermo Bistue and Angel Rubio**

Engineering School TECNUN, University of Navarra, Paseo de Manuel Lardizabal, 13,
20018 Donostia San Sebastian, Spain; gbistue@tecnun.es (G.B.); arubio@tecnun.es (A.R.)
\* Correspondence: amedina@tecnun.es; Tel.: +34-943-219877

**Abstract:** Direct Yaw Moment Control (DYC) is an effective way to alter the behaviour of electric cars with independent drives. Controlling the torque applied to each wheel can improve the handling performance of a vehicle making it safer and faster on a race track. The state-of-the-art literature covers the comparison of various controllers (PID, LPV, LQR, SMC, etc.) using ISO manoeuvres. However, a more advanced comparison of the important characteristics of the controllers' performance is lacking, such as the robustness of the controllers under changes in the vehicle model, steering behaviour, use of the friction circle, and, ultimately, lap time on a track. In this study, we have compared the controllers according to some of the aforementioned parameters on a modelled race car. Interestingly, best lap times are not provided by perfect neutral or close-to-neutral behaviour of the vehicle, but rather by allowing certain deviations from the target yaw rate. In addition, a modified Proportional Integral Derivative (PID) controller showed that its performance is comparable to other more complex control techniques such as Model Predictive Control (MPC).

**Keywords:** direct yaw moment control; electric race car; FSAE; limit handling; yaw rate control; lap time simulation





## 1. Introduction

The irruption of electric technology in the automotive industry is setting a new and important milestone in automotive history. The proliferation of electric vehicles in the coming years seems clear, not only by looking at the different roadmaps but also by considering the necessities that are arising from the social, economic, mobility, and ecology fields. Electric technology brings also several advantages to the field of vehicle dynamics control. The ability (and ease) of installing an electric motor in every single wheel introduces another degree of freedom in terms of vehicle handling [1–3]. Apart from this, the short response time of the electric motors provides a more effective transmission of the motor torque to the wheel [4]. Direct Yaw Moment Control (DYC) systems, also recently denominated as Torque Vectoring (TV), take advantage of these benefits on vehicles with independent motor configurations, as motor torque can be distributed independently on each wheel. In the last three decades, DYC has been intensively investigated. Many contributions have been proposed to the employment of different control methods for DYC, especially in the last ten years, given the torque distribution freedom of electric powertrains with independent motors, but also in some cases due to the advancements made in electronic differentials. Control methods such as PID (Proportional Integral Derivative), LPV (Linear Parameter-Varying), LQR (Linear Quadratic Regulator), LQG (Linear Quadratic Gaussian), H-infinity, Fuzzy Logic, SMC (Sliding Mode Control), and MPC (Model Predictive Control) have been investigated in recent years, some of them with a combination of feedforward techniques. Typical control variables in such controllers are yaw rate and sideslip angle.

PID controllers regulate based on the error, the derivative, and the integral of the error between a reference and an actual value. PID controllers have two main advantages.

The first one is the ease of implementation resulting in easier tuning and minimal computational requirements. Secondly, PID controllers are known to be relatively robust, i.e., able to withstand changes in the vehicle modelling without compromising the stability of the vehicle. There are also some drawbacks: because of its simplicity, a PID controller cannot exploit a complete knowledge of the vehicle. If the plant is perfectly identified, a PID controller is outperformed by other algorithms. PID are quite spread as yaw rate controllers. Table 1 shows several studies using PID controllers as main control algorithms [5–15].

**Table 1.** Proportional Integral Derivative (PID) controllers in the literature on Direct Yaw Moment Control (DYC).

| Ref(s). | FF [1] Terms | Comments |
|---|---|---|
| [5] | No | Reduces slip angle difference between front and rear axle to achieve maximum lateral acceleration. |
| [6] | Yes | Focuses on rear cornering stiffness to avoid instability, evaluates control using a driving simulator. |
| [7] | Yes | Derives an analytical formula to improve the steady and transient dynamics of the vehicle. |
| [8] | Yes | Minimizes sideslip angle. |
| [9] | Yes | Minimizes yaw rate error between a reference model and the real vehicle. |
| [10] | Yes | Combined with active front steering. |
| [11] | No | Reference tracking and proposes a tuning method. Tested on the ISO 3888-2 Double Lane Change Test at 40 km/h and 90 km/h. |
| [12] | Yes | Estimates sideslip angle and cornering stiffness through a Kalman filter. Compared vs. friction brake actuation. |
| [13] | Yes | Wheel torque distribution criteria using offline optimization and Control Allocation (CA). |
| [14] | Yes | Performance comparison with H-infinity controller. |
| [15] | No | Uses a cubic-error PD controller for yaw rate and sideslip control. |

[1] FF: feed Forward terms.

Sliding Mode Control (SMC) uses an arbitrarily large gain to force the behaviour of a dynamic system to follow a trajectory of a reduced-order system (usually order one or two). The main strength is robustness against modelling uncertainties. From the point of the drawbacks, the controller is usually extremely active (the actuators are continuously saturated), which in turn provokes chattering. Dead-band controllers, low-pass filters, or integral actuation can improve this problem. Table 2 summarizes the main publications on SMC controllers related to DYC [16–25].

Linear Quadratic Regulators (LQR) are optimal controllers that balance the tracking performance of the state variables (minimization of the overall error of the yaw rate) with the actuation (commanded asymmetrical torque on the wheels). The simplest LQR controllers minimize the integral of a weighted sum of the squared error and the square of the actuation. The gains for these types of controllers can be obtained by solving the corresponding Ricatti equation [26]. LQR is an optimal controller and the proper selection of the function to minimize provides very effective results. On the downside, they are very sensitive to mismodelling the vehicle or perturbations and are usually applied to linear systems. Linear Quadratic Gaussian (LQG) is a variant of LQR that also includes disturbances (e.g., side winds) modelled as Gaussian noise. If properly tuned, the effect of disturbances can be minimized. Table 3 shows the literature related to optimal controllers (both LQR and LQG) for DYC [27–35].

**Table 2.** Sliding Mode Control (SMC) controllers in the literature on DYC.

| Ref(s). | Order [1] | Comments |
|---------|-----------|----------|
| [16] | 2 | Two second-order sliding-mode controllers are evaluated against a feedforward controller combined with either a conventional or an adaptive Proportional Integral Derivative (PID) controller. |
| [17] | −1 | Implements Integral Sliding Mode Control (ISMC) to avoid chattering |
| [18] | 1 | Combines SMC with PID. Include a low pass filter to reduce chattering. Reduces the difference between front and rear slip angles. |
| [19] | 1 | Includes saturation to reduce chattering. |
| [20] | −1, 2 | Compares Internal Mode Control (IMC) and Second-Order Sliding Mode Control (SOSM), both using feedforward terms. With both control techniques, stability in demanding oversteering conditions, such as braking in a high-speed turn, can be worse than the uncontrolled case, depending on the longitudinal deceleration level. |
| [21] | 1 | Multiple Adaptive Sliding Mode Control (MASMC). |
| [22] | - | Compares Integral Terminal Sliding Mode (ITSM) and Nonsingular Fast Terminal Sliding Model (NFTSM) to improve the transient response of the vehicle sideslip angle and yaw rate. |
| [23] | 1 | Adaptive. Lyapunov-based stability analysis. Performance studied on a double lane change test simulation. |
| [24] | 1, 2 | Compares first order, SOSM, and PID controllers. SOSM is the winner of the comparison based on a Sine with Dwell test manoeuvre (no chattering, best tracking performance, better slip-angle). |
| [25] | −1 | Integral Sliding Mode Control (ISMC) compared LQR controller with and without non-linear feedforward. ISMC outperforms LQR both in tracking performance and yaw damping. |

[1] Order is 1 for first order SMC, 2 for Second-Order Sliding Mode Control (SOSM) and −1 for Integral Sliding Control.

**Table 3.** Optimal controllers applied to DYC.

| Ref(s) | Controller | Comments |
|--------|------------|----------|
| [27,28] | LQR | Applied to production vehicles. |
| [29] | LQR | Tracks vehicle yaw rate, minimization of the optimal handling performance index. |
| [30] | LQR | Tracks yaw rate and sideslip angle, minimizes the use of external yaw moment. |
| [31] | RLQR | Robust controller. Robustness is achieved through gain-scheduling and additional closed-loop control terms. Outperforms standard LQR. |
| [32] | LQR | Applied to Formula Student vehicle. Showed promising results compared to PD controller. |
| [33,34] | LQG | Improved disturbance rejection ability if compared to LQR. |
| [35] | LPV | DYC combined with torque and slip limitation applied to a front-wheel-drive electric vehicle. |

LQR and LQG were developed to control linear systems. A vehicle is not linear (especially at handling limits) and the application of these controllers requires the linearization of the vehicle for a particular working point. MPC, discussed later, tries to fix this problem by solving an optimization equation online. MPC are more computationally demanding compared to LQR and LQG controllers. Another possible approach is to use a gain scheduling method as performed in linear parameter-varying controllers (LPV).

Fuzzy logic controllers have also been applied to DYC. They consist of three main parts: fuzzification, rule processor, and defuzzification. The rules are usually stated by the control designer intuitively; e.g., "if the yaw error is large, apply a large torque to diminish it". The fuzzification part converts the input measurements into qualitative quantities; i.e., state if a specific yaw error is "large" or "huge". Usually the regions between the qualitative measurements ("large" and "huge") overlap. The defuzzification consists of generating the

specific control action according to the output rules; e.g., a large torque is at least 60 Nm. These three parts of the controllers (fuzzification, generation of rules, and defuzzification) require an in-depth knowledge of the process under study. Nevertheless, fuzzy controllers have been successfully applied to DYC or even to unstable systems (Table 4) [36–39].

**Table 4.** Fuzzy controllers applied to DYC.

| Ref(s) | Comments |
|---|---|
| [36] | A high-level supervisory module operated by a genetic fuzzy yaw moment controller. |
| [37] | Comparison to an LQR. Fuzzy logic shows better results on ISO3888-2 and Sine with Dwell manoeuvres. |
| [38] | A unified controller with three control layers based on fuzzy control strategy is designed for this purpose and applied on a vehicle with an electronic differential. |
| [39] | A neuro-fuzzy vertical tire forces estimator combined with a fuzzy yaw moment controller is compared to a more traditional PID controller using a high-fidelity vehicle dynamics simulator; results show that the proposed controller can increase vehicle efficiency by 10%. |

Model Predictive Control (MPC) is similar to LQR controllers with some key differences. They are similar as long as both solve an optimization problem that trades off the tracking ability and the actuation. However, the approach is different since the optimization problem is solved online, with the additional computational cost, and can include non-linearities (such as actuator saturation) as long as the optimization solver can deal with them. Moreover, the optimization problem is solved for a finite-time horizon; i.e., MPC minimizes a figure of merit that includes some samples, not an infinity summation as in LQR. Table 5 includes a summary of the DYC controllers that use the MPC approach [40–51].

**Table 5.** Model Predictive Control (MPC) applied to DYC.

| Ref(s) | Controller | Comments |
|---|---|---|
| [40] | Non-linear | Nearest point approach. Applied to step steer and split braking manoeuvre. |
| [41] | Standard | Applied to U-turn and double lane change. Outperforms LQR. |
| [42] | Non-linear | Robust controller. Robustness achieved using gain-Model In combination with an SMC to compute the necessary torques on the rear wheels based on the requested longitudinal slips. Outperforms LQR. |
| [43] | Standard | The linear vehicle model is used for the MPC and compared with an equal torque algorithm. Evaluation is performed by simulation. |
| [44] | Adapted to deal with delay | Yaw response of the vehicle is improved through torque vectoring to track the desired yaw rate, even with the presence of delays in the control loop which could degrade controller performance. Effectiveness is verified by simulation and by experiments with a rear-wheel-drive electric vehicle |
| [45] | 2 controllers: Standard and non-linear | Applied to Formula Student car. Both use the qpOASES solver [46]. The nonlinear model uses ACADO code generation tool [47]. Tested for U-turn and step steer. |
| [48] | Standard | Requires no road friction information. Estimated using the relative difference between front and rear slip angles. |
| [49] | Non-linear | Both torque vectoring and Electronic Stability Control (ESC). Non-linearity includes constraints in the actuators. Tested on line-change and J-turn manoeuvres. |

**Table 5.** *Cont.*

| Ref(s) | Controller | Comments |
|--------|-----------|----------|
| [50] | Standard with physical constraints | Applied to 4WD. Tested on step steer and double lane-change manoeuvres. Outperforms LQR. |
| [51] | Non-linear | Concurrent optimization of the reference yaw rate and wheel torque allocation. Cost function weights on-line varied using fuzzy logic to adaptively prioritize vehicle dynamics or energy efficiency. |

In most of the previous works, performance evaluation was performed by comparing specific manoeuvres (ISO 3888-2 Double Lane Change, Sine with Dwell test, etc.), but only a few were evaluated on a complete race track. In a race car, ISO 3888-2 or other manoeuvres have little or no importance at all: the critical ones are the lap time and the robustness of the method under different circumstances (wet asphalt, a significant change of the aerodynamic behaviour because of a crash, etc.)

Besides, a comprehensive and systematic comparison of different control techniques for DYC is missing in the literature, to the knowledge of the authors. This manuscript will perform a systematic comparison of the main control approaches (PID, LQR, SMC, MPC, and NMPC) and their performance is evaluated on two race tracks in this study.

These studies have been performed using the IPG CarMaker software using an "expert driver model", since a race car is being studied. The vehicle powertrain and the controllers have been programmed in MATLAB Simulink, which is connected to IPG CarMaker.

Results show the difference in actuation for each controller, as well as the impact on yaw and sideslip dynamics and finally on the lap time. Interestingly, there are some counterintuitive conclusions in the results: best lap times are not achieved using a neutral behaviour. On the other hand, PID controllers withstand the comparison with other control techniques.

## 2. Vehicle Modelling

### 2.1. Vehicle Specifications

A two-wheel-drive Formula Student vehicle is considered with two rear independent motors which can be controlled individually. The total power output of the battery is limited to 80 kW, according to the competition rules. Each motor can deliver a peak torque of 90 Nm, which is translated into 450 Nm of torque at each wheel owing to the 1:5 gear reduction factor. The vehicle features a complete aerodynamic package with a rear-biased downforce which increases its understeering behaviour with the rising speed. Stiff chassis, a low centre of gravity, and small vehicle mass and inertia, in combination with high grip tyres, allows very good handling behaviour in terms of yaw rate response, also at combined acceleration.

### 2.2. Linearized Bicycle Model

A linearized bicycle model [52,53] is considered to describe the yaw and sideslip dynamics of the vehicle, as shown in Figure 1.

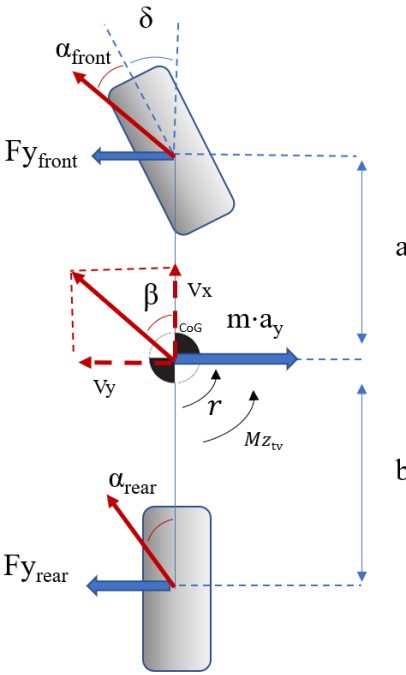

**Figure 1.** Linearized bicycle model.

Yaw and lateral motions are defined as follows:

$$\sum F_y = F_{y,front} + F_{y,rear} = mv_x\left(\dot{\beta} + r\right) \tag{1}$$

$$\sum M_z = F_{y,front}a - F_{y,rear}b + M_{z,tv} = I_z\dot{r} \tag{2}$$

Assuming linear tyres around the operating point and their respective cornering stiffnesses $C_{\alpha,front}$ and $C_{\alpha,rear}$ and relating front and rear slip angles $\alpha_{front}$ and $\alpha_{rear}$ to vehicle sideslip angle $\beta$, yaw rate $r$, and front-wheel steer angle $\delta$, we have:

$$F_{y,front} = \alpha_{front}C_{\alpha,front} \quad F_{y,rear} = \alpha_{rear} \cdot C_{\alpha,rear} \quad \alpha_{front} = \beta + \frac{r \cdot a}{v_x} - \delta \quad \alpha_{rear} = \beta - \frac{r \cdot b}{v_x}$$

$$\dot{\beta} = \frac{C_{\alpha,front} + C_{\alpha,rear}}{m \cdot v_x} \cdot \beta + \left(\frac{C_{\alpha,front} \cdot a - C_{\alpha,rear} \cdot b}{m \cdot v_x^2} - 1\right) \cdot r - \frac{C_{\alpha,front}}{m \cdot v_x} \cdot \delta \tag{3}$$

$$\dot{r} = \frac{C_{\alpha,front} \cdot a - C_{\alpha,rear} \cdot b}{I_z} \cdot \beta + \frac{C_{\alpha,front} \cdot a^2 + C_{\alpha,rear} \cdot b^2}{I_z \cdot v_x} \cdot r - \frac{C_{\alpha,front} \cdot a}{I_z} \cdot \delta + \frac{1}{I_z} \cdot M_{z,tv} \tag{4}$$

Equations (3) and (4) can be expressed in state-space representation form:

$$\dot{x} = A \cdot x + B \cdot u \tag{5}$$

Being:

$$x = \begin{pmatrix} \beta \\ r \end{pmatrix} \qquad A = \begin{pmatrix} a_{11} & a_{12} \\ a_{21} & a_{22} \end{pmatrix} \qquad B = \begin{pmatrix} b_{11} & b_{12} \\ b_{21} & b_{22} \end{pmatrix} \qquad u = \begin{pmatrix} \delta \\ M_{z,tv} \end{pmatrix}$$

$$a_{11} = \frac{C_{\alpha,front} + C_{\alpha,rear}}{m \cdot v_x} \qquad a_{12} = \frac{C_{\alpha,front} \cdot a - C_{\alpha,rear} \cdot b}{m \cdot v_x^2} - 1 \qquad a_{21} = d \frac{C_{\alpha,front} \cdot a - C_{\alpha,rear} \cdot b}{I_z}$$

$$a_{22} = \frac{C_{\alpha,front} \cdot a^2 + C_{\alpha,rear} \cdot b^2}{I_z \cdot v_x} \qquad b_{11} = -\frac{C_{\alpha,front}}{m \cdot v_x} \quad b_{12} = 0 \qquad b_{21} = -\frac{C_{\alpha,front} \cdot a}{I_z} \quad b_{22} = \frac{1}{I_z}$$

Bicycle model properties, including both axles cornering stiffness change over the vehicle speed range, are summarized in Table 6. Yaw rate reference for all the controllers is based on the yaw rate response to front-wheel steer angle input transfer function derived from the state-space equation. For simplicity, and due to the good handling properties of the passive vehicle, yaw rate reference is approximated by a first-order transfer function, as exposed in [15]. State-space matrixes are used to calculate the optimal gain matrix for LQR and also as an internal vehicle model for the MPC controllers.

**Table 6.** Linearized bicycle model properties.

| Parameter | Value |
|---|---|
| Vehicle mass (driver included), $m$ | 296 kg |
| Yaw Inertia, $Iz$ | 153 kg m$^2$ |
| Wheelbase, $L$ | 1.58 m |
| Distance from the front axle to the centre of gravity, $a$ | 0.798 m |
| Distance from the rear axle to the centre of gravity, $b$ | 0.782 m |
| Front axle cornering stiffness, $C_{\alpha,front}$ (absolute value) | |
| At 20 km/h | 37,530 N/rad |
| At 40 km/h | 42,660 N/rad |
| At 60 km/h | 47,780 N/rad |
| At 80 km/h | 52,900 N/rad |
| At 100 km/h | 58,000 N/rad |
| Rear axle cornering stiffness, $C_{\alpha,rear}$ (absolute value) | |
| At 20 km/h | 39,400 N/rad |
| At 40 km/h | 49,100 N/rad |
| At 60 km/h | 58,800 N/rad |
| At 80 km/h | 68,500 N/rad |
| At 100 km/h | 78,200 N/rad |

## 3. Controllers Description

The general scheme for the controllers is shown in Figure 2. Baseline vehicle characteristics are calculated from the state space equations and translated into yaw rate gain $Y_g$ and yaw rate delay time constant $\tau$ for the first order simplified vehicle response. The target yaw rate $r_{target}$ is bounded according to the vehicle's maximum cornering capability considering aerodynamic downforce and tarmac grip. $r_{target}$ is then compared to actual yaw rate $r$, and the difference is sent to the yaw rate controller. The controller will then compute the necessary corrective yaw moment $Mz_{tv}$ and convert it into throttle signals to be combined with the actual longitudinal force command coming from the driver gas pedal. Left and right throttle signals are bounded to $[-1,1]$, the maximum electric motor commands in a back-and-forth direction.

Before sending the throttle command to the motors, the signal passes through a supervisor traction controller which limits the torque command according to the maximum longitudinal force capability of the tire.

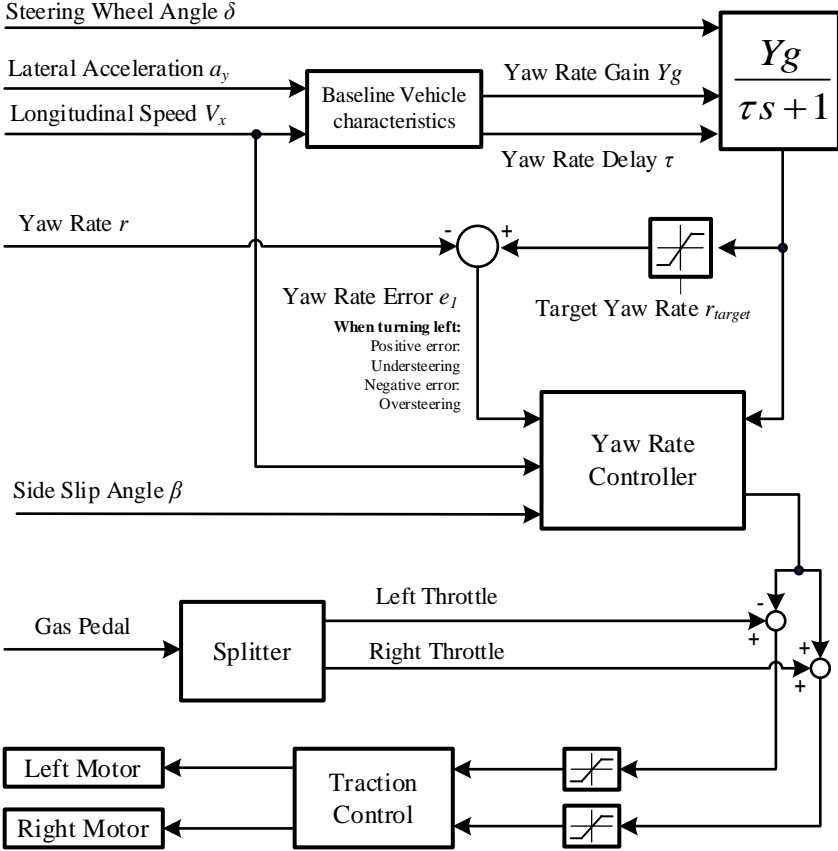

**Figure 2.** General controller scheme.

### 3.1. PID

An error-cubic PD controller introduced in [15] is used. This controller applies a non-linear transform of the error prior to the application of the proportional and derivative gains. The gains are chosen according to the minimum lap time achieved in the simulations. The overall arrangement is shown in Figure 3.

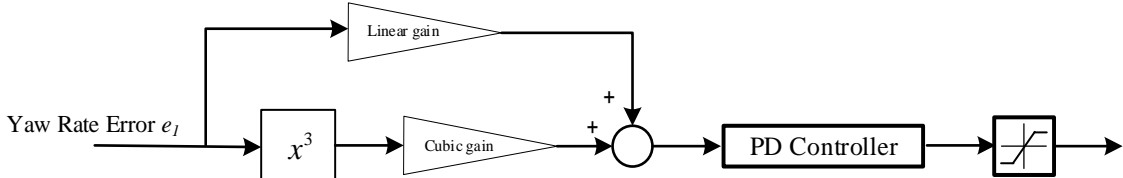

**Figure 3.** Error-cubic PD controller.

### 3.2. Sliding Mode Controller

A first-order SMC controller which is based on [19] is employed. A low pass filter is added on top of the saturation function to further eliminate control chattering and smoothen the control output. The gains are chosen by iteration and minimum lap time in the simulations. The overall scheme is shown in Figure 4.

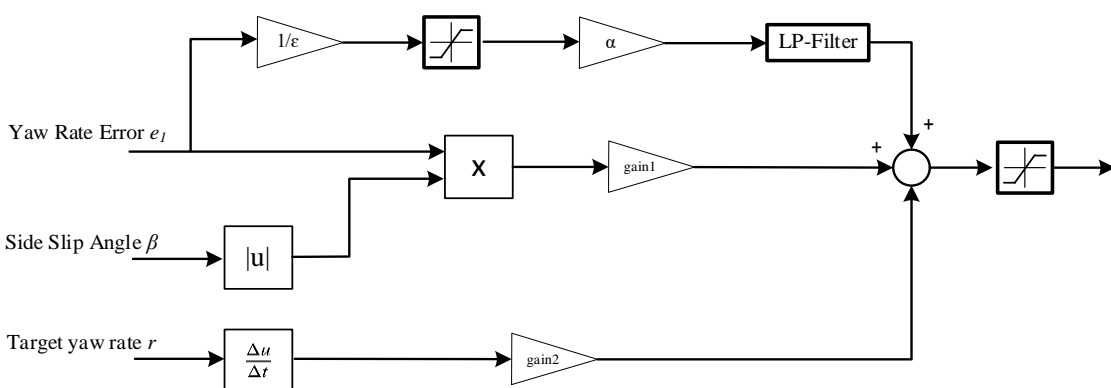

**Figure 4.** Sliding mode controller with a low-pass filter.

### 3.3. Linear Quadratic Regulator

The LQR gains are calculated by the state-space representation coefficients for the speed range mentioned in Section 2. The optimal gain matrix K is calculated using Matlab *lqr* function. The state feedback law $u = -k \cdot x$ minimizes the following cost function [54]:

$$J(u) = \int_0^\infty \left( x^T \cdot Q \cdot x + u^T \cdot R \cdot u + 2 \cdot x^T \cdot N \cdot u \right) \cdot dt \tag{6}$$

The chosen weight matrixes being:

$$Q = 1e7 \cdot \begin{pmatrix} 0 & 0 \\ 0 & 1 \end{pmatrix} \cdots\cdots R = 1 \cdots\cdots N = 0$$

It can be observed that the chosen weight on the sideslip angle is zero, and the yaw rate error is much more penalized than actuation. A function obtained from the fitting of these gains over speed is calculated. A dead zone function is added to the yaw rate error to prevent the controller actuation for small yaw rate errors, as shown in Figure 5.

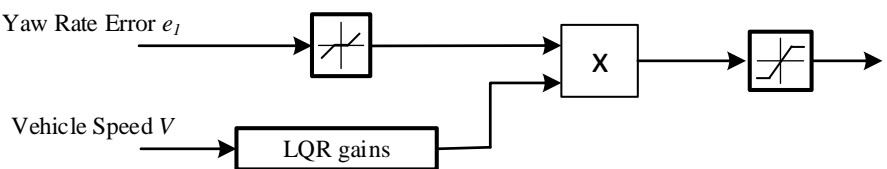

**Figure 5.** Linear Quadratic Regulator controller scheme.

### 3.4. Linear Model Predictive Control

The following linear MPC formulation is considered:

$$\min_{U,X} \sum_{i=0}^{n_p-1} \| x_{k+i} - x_{ref} \|_Q^2 + \| u_{k+i} \|_R^2 \tag{7}$$

Subject to:

$x_k = \hat{x}(k)$
$x_{k+i+1} = A_d \cdot x_{k+i} + B_d \cdot u_{k+i}$

$-2138 \leq M_{z,tv} \leq 2138 \ [\text{Nm}]$

$i = 0, 1, \ldots, n_p - 1$

$A_d$ and $B_d$ being the discrete state space matrixes of (5) with a time discretization of $T_d = 0.01$ seconds. These matrixes update every time step with the vehicle speed, as shown

in Figure 6. A horizon of $n_p = 40$ steps. For the algorithm implementation, ACADO Code Generation Tool [47] is used.

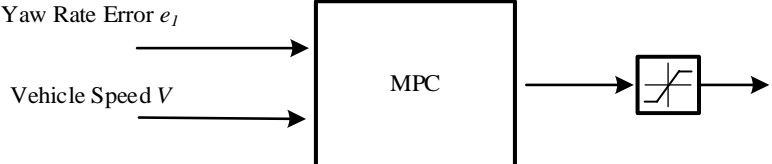

**Figure 6.** Model Predictive Control scheme.

### 3.5. Linear Parameter Varying Model Predictive Control

Another MPC controller is used with similar characteristics to the previous one, but this time an online linearization of the cornering stiffness of both axles is performed for a more accurate internal model. Figure 7 illustrates the control scheme.

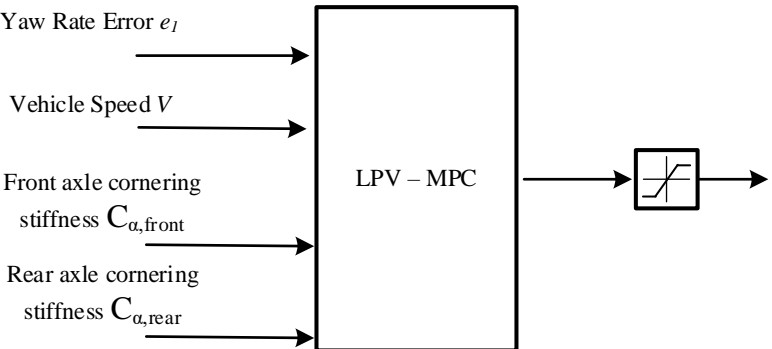

**Figure 7.** Linear Parameter Varying Model Predictive Control scheme.

## 4. Simulation Model

### 4.1. Vehicle Parameters

Simulations are performed using IPG CarMaker. The full vehicle model is parametrized, including chassis bending and torsional stiffnesses. Tyres are parametrized using Magic Formula 5.2 [53], including the relaxation length. Sprung and unsprung mass and inertia, suspension kinematics, and aerodynamic map are also included. Controllers and powertrain and modelled on Matlab Simulink, which runs on co-simulation with IPG CarMaker. Two track models are used for the simulations: a go-kart track [55] and the Formula Student Germany 2010 Endurance track [56] (included in the IPG Formula CarMaker sponsorship programme). The driver is characterized using IPG Racing Driver model. The main parameters for the driver model are the learning rate and the driver combined acceleration target.

### 4.2. Olaberria Circuit

Circuito de Olaberria is an existing go-kart track located in northern Spain. Two chicanes, two 180 degrees corners, and one delicate braking point after a high-speed left corner distinguish this track, which combines tight corners, which can be similar to the ones that can be found on a Formula Student Endurance event and relatively high-speed areas that are very exigent to the vehicle and driver and exploit the aerodynamic package potential of the vehicles.

The track in this chosen configuration (pathway can be altered for different configurations) is 695 m long. Chosen coefficient of friction is $\mu = 1$. The modelling of some patches of the tarmac is also included, with a coefficient of friction of $\mu = 0.9$. The track layout is described in Figure 8.

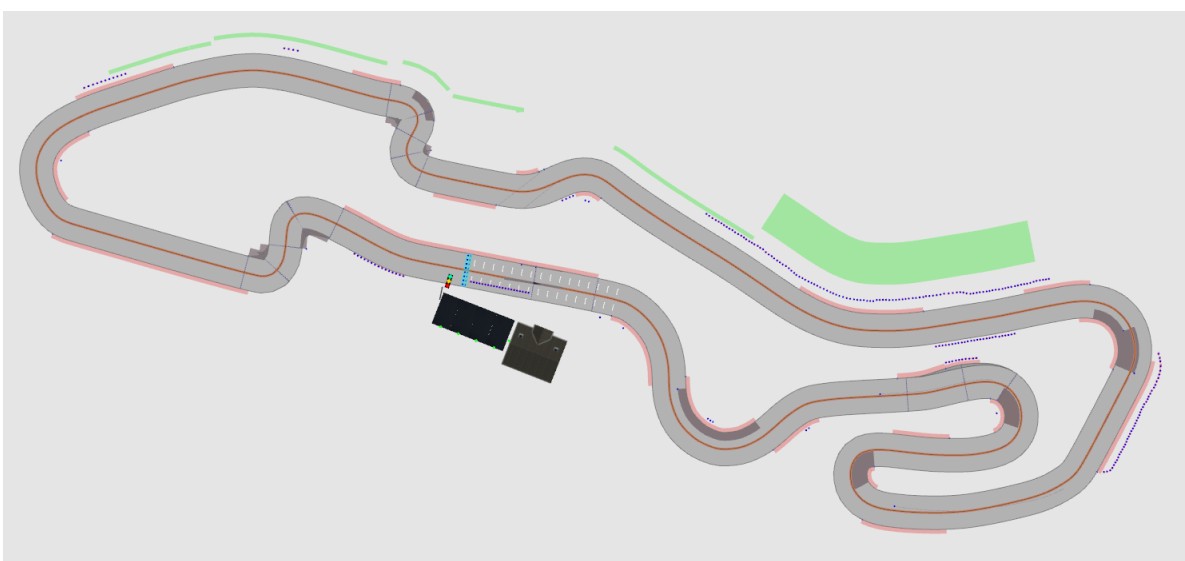

**Figure 8.** Olaberria circuit aerial view.

### 4.3. Hockenheim 2010 Formula Student Endurance Track

The Endurance track of the 2010 Formula Student Germany event is also employed for the simulations, taking advantage of fact that this track is already modelled and included in the IPG Formula CarMaker sponsorship programme. The highlights of this track are multiple medium speed linked corners (which are typical in this kind of Formula Student Endurance events), one long straight on which typically the vehicles achieve their maximum speed, and three long right corners. The length of this track is 774 m and, lap times are higher than in Olaberria, due to the longer track length and the amount of aforementioned linked medium speed corners. A coefficient of friction μ = 1 has been set for the road grip. Track aerial view is shown in Figure 9.

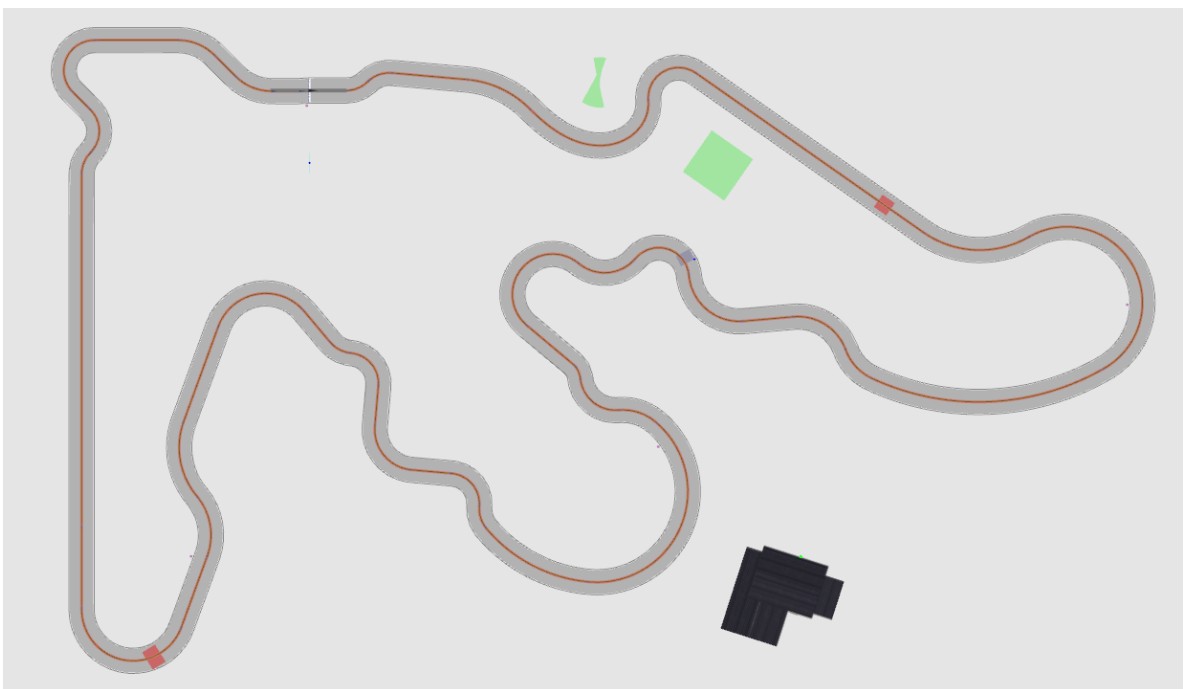

**Figure 9.** Hockenheim 2010 Formula Student Endurance track aerial view.

## 5. Results

Several simulations were performed on IPG-CarMaker for each controller configuration to find the quickest lap time. For this, an iterative process is required, varying the learning rate and the driver target for combined acceleration. Best times are usually achieved at high learning rates and combined acceleration usage targets. Besides lap time, two more key performance indicators have been calculated to evaluate controller actuation:

- Integral of the absolute yaw rate error (IAE):

$$IAE = \int_0^{t_{lap}} \| e(t) \| \, dt \qquad (8)$$

- Integral of the absolute value of the control action (IACA):

$$IACA = \int_0^{t_{lap}} \| u(t) \| \, dt \qquad (9)$$

Lap times at the Olaberria circuit are shown in Table 7. As can be observed on the table, the best lap times are achieved by the most aggressive driving style and highest combined acceleration target. Higher driver target G-G exponents lead to vehicle loss of control due to front wheel lock with heavy braking (there is no ABS (Anti Lock Braking System) in the vehicle [57]).

**Table 7.** Lap times at Olaberria.

| Race Driver Parameters | | Lap Times | | | | |
|---|---|---|---|---|---|---|
| Learning Rate: 0: Sensitive 1.5: Aggressive | Driver Target G-G Exponent | PID | SMC | LQR | MPC | LPV-MPC |
| 0 | 1 | 36.92 | 36.94 | 36.73 | 36.78 | 36.80 |
| 0.5 | 1.2 | 36.13 | 36.26 | 35.97 | 35.99 | 36.07 |
| 0.7 | 1.4 | 35.68 | 35.78 | 35.50 | 35.55 | 35.62 |
| 1 | 1.6 | 35.33 | 35.48 | 35.14 | 35.28 | 35.32 |
| 1.5 | 1.8 | 35.12 | 35.42 | 34.98 | 35.16 | 35.11 |
| 1.5 | 2 | 35.1 | 35.27 | **34.80** | 35.13 | 35.00 |
| 1.5 | 2.2 | **34.85** | **35.23** | 34.82 | **34.82** | **34.78** |
| 1.5 | 2.3 | DNF | DNF | DNF | DNF | DNF |

DNF = Do not finish. Numbers in bold indicate fastest lap for each controller.

Among the controllers, the LPV-MPC is the one that achieves the fastest lap time. The PID intervenes less than the other controllers by looking at the IACA values. The MPC, on the other hand, keeps the error between the target and the actual yaw rate to the minimum, but at the expense of a busy controller intervention, as can be seen in Table 8.

**Table 8.** Controller performance at Olaberria.

| Controller | Best Lap Olaberria | | |
|---|---|---|---|
| | Lap Time (s) | IACA (Nm·s) | IAE (rad) |
| PID | 34.85 | **3103** | 7.34 |
| SMC | 35.23 | 6660 | 7.70 |
| LQR | 34.80 | 6867 | 6.77 |
| MPC | 34.82 | 8105 | **5.63** |
| LPV-MPC | **34.78** | 5130 | 8.39 |

Regarding the simulation results at Hockenheim circuit, the best lap times are also achieved by the maximum value of learning rate and maximum driver target G-G exponent, as can be observed in Table 9. However, there is a difference with Olaberria track

simulations: in Hockenheim, the driver never loses vehicle control. There are two reasons for this:

- The braking locations at Hockenheim require less combined G (the vehicleheads straight when braking).
- Vehicle speed at the braking areas at Hockenheim is lower than in the critical braking point of Olaberria, where the front axle lock occurs. At lower speed, the ratio of apparent front wheels weight over the apparent weight on the rear wheels is higher, so it is more difficult to lock the front wheels.

**Table 9.** Lap times at Hockenheim 2010 Formula Student.

| Race Driver Parameters | | Lap Times (s) | | | | |
|---|---|---|---|---|---|---|
| Learning Rate: 0: Sensitive 1.5: Aggressive | Driver Target G-G Exponent | PID | SMC | LQR | MPC | LPV-MPC |
| 0 | 1 | 49.97 | 49.96 | 49.96 | 49.97 | 49.95 |
| 0.5 | 1.2 | 49.28 | 49.19 | 49.30 | 49.31 | 49.21 |
| 0.7 | 1.4 | 48.84 | 48.73 | 48.90 | 48.89 | 48.71 |
| 1 | 1.6 | 48.46 | 48.36 | 48.53 | 48.51 | 48.37 |
| 1.5 | 1.8 | 48.28 | 48.24 | 48.38 | 48.36 | 48.20 |
| 1.5 | 2 | 48.14 | 48.07 | 48.19 | 48.17 | 48.09 |
| 1.5 | 2.2 | 48.06 | 47.95 | 48.12 | 48.08 | 48.01 |
| 1.5 | ∞ | **47.34** | **47.31** | **47.41** | **47.38** | **47.28** |

Nevertheless, the combined G-G usage during vehicle acceleration is controlled by the torque vectoring and traction control system, so no instability happens during heavy longitudinal accelerations at a corner exit. If these controllers are removed, the vehicle spins at a corner exit when targeting high combined acceleration.

With regards to controller comparison, the LPV-MPC achieves again the best lap, but the differences among them in these terms are tiny. PID is again the least intrusive controller, and the MPC again achieves the best tracking performance at the cost of having the highest intervention, as indicated on Table 10.

**Table 10.** Controller performance at Hockenheim.

| Controller | Best Lap Hockenheim | | |
|---|---|---|---|
| | Lap Time (s) | IACA (Nm·s) | IAE (rad) |
| PID | 47.34 | **2499** | 4.69 |
| SMC | 47.31 | 5414 | 5.29 |
| LQR | 47.41 | 3881 | 4.22 |
| MPC | 47.38 | 5483 | **4.02** |
| LPV-MPC | **47.28** | 3229 | 5.90 |

## 6. Discussion

This work compares different controller architectures for torque vectoring. Instead of using ISO manoeuvres, they are compared using lap times in two different circuits. Simulations were performed using IPG CarMaker software, with a Formula Student 2WD vehicle model on two race tracks, using IPG Race Driver model.

The difference in lap times across different controllers is tiny: the best lap times for different controllers are within the same tenth of a second, except for SMC which is the slowest in Olaberria. Nonetheless, there are interesting differences across the controllers.

The most important factor is the driver: aggressive driving with a large G-G exponent provides faster laps. In Olaberria, tough, hard braking at the fastest point of the circuit provokes front axle locking, and if the G-G exponent is over 2.2 the car is not able to finish the lap with any controller.

Interestingly, the error in the yaw rate is not critical for lap times: both in Hockenheim and Olaberria, MPC and LPV-MPC have the largest and smallest IAE, and the corresponding lap times are within the same tenth. Interestingly, MPC, whose IAE is smaller, is in fact slower in both circuits than LPV-MPC. This suggests that in order to extract the maximum tyre force, the target yaw rate function could be improved. A comparison between a "fast" and a "slow" lap can be seen in Figure 10.

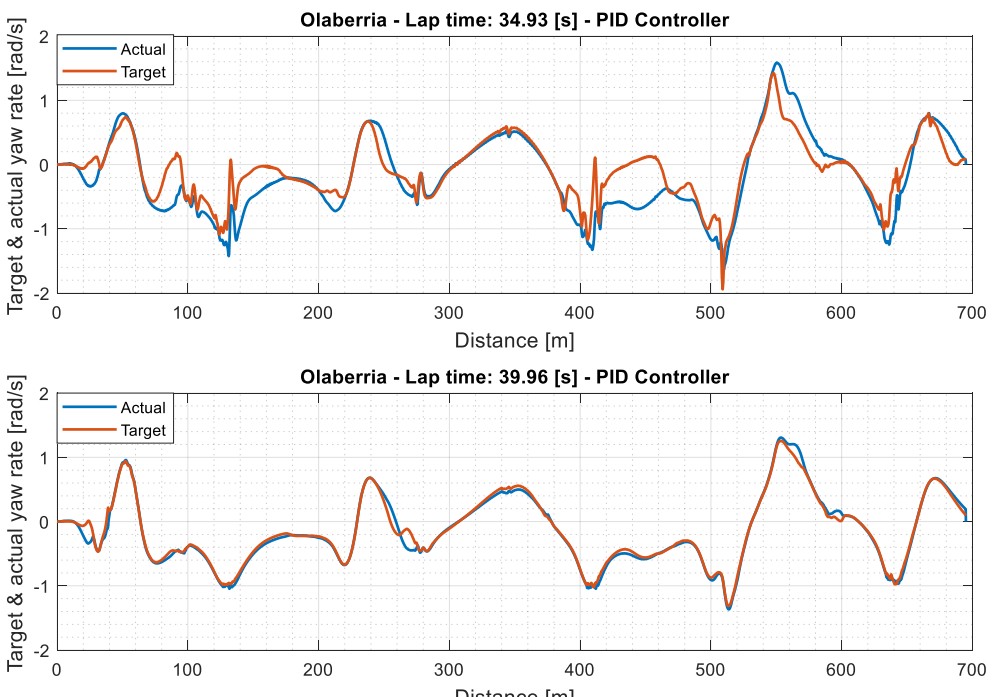

**Figure 10.** Target vs. actual yaw rate in a "fast" and a "slow" lap.

It can be observed that the "slow" lap has a much smaller IAE value than the "fast" lap. The reason is that the slip angle of the maximum lateral force of the tyre varies constantly during demanding driving. The target yaw rate is based on a fixed relation between the front and rear slip angles; it is constrained to this relation and does not take into account the exact slip angle for maximum lateral performance of the tyre (it simply assumes that both axles slip angles need to be always similar, or the front should be always slightly higher for understeering behaviour).

Tables 8 and 10, and Figure 11 comparethe control action of each controller. In this case, the differences are much clearer than in lap times. The PID controller is much less intrusive than the others are, and its lap time is less than one-tenth slower than the LPV-MPC in each circuit. Despite its simplicity, the performance of the PID controller is remarkable.

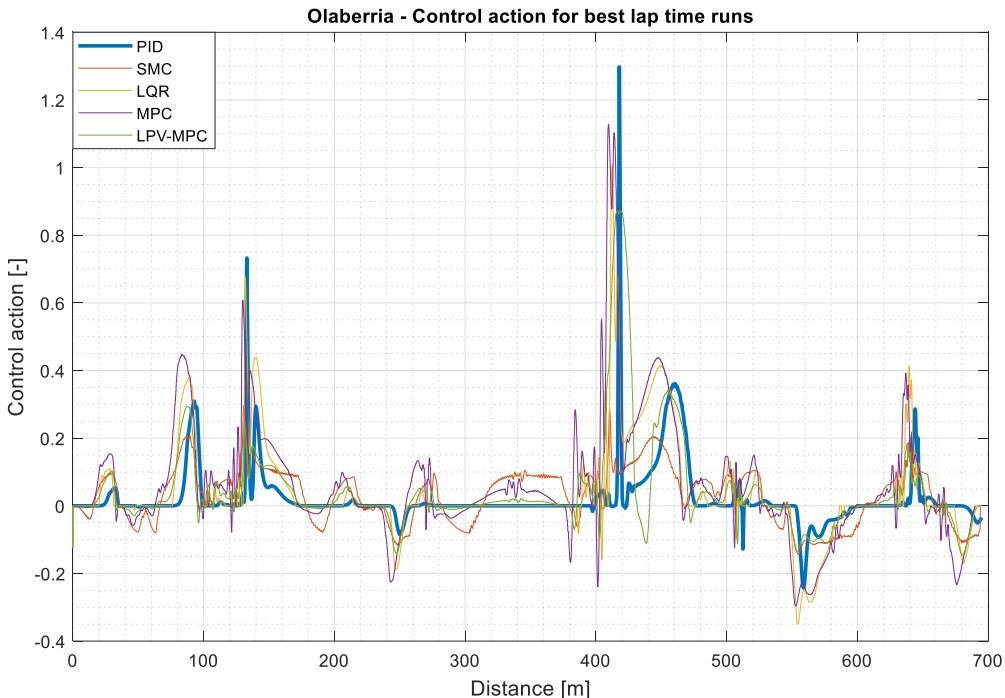

**Figure 11.** Control action of each controller for best lap time runs.

## 7. Conclusions

A review of the typical controllers for direct yaw moment control applied on an electric race car with independent motors was performed. The comparison shows that all controllers perform similarly in terms of minimum lap time according to the simulations performed on IPG CarMaker. However, there are some differences in terms of controller actuation: the cubic-error PID is the least intrusive controller, whereas the MPC controller shows the best yaw rate tracking performance. Nevertheless, tracking performance is not directly correlated with minimum lap time, due to the continuous variation of the tyre peak slip angles during race-style driving. Therefore, future works include the design of a target yaw rate function considering the peak performance of each tyre. Besides this, the evaluation of the controllers using a moving base driving simulator is also considered to evaluate the intrusiveness and the interaction with a human driver.

**Author Contributions:** A.M.: Conceptualization, formal analysis, writing and editing. G.B.: Supervision. A.R.: Supervision, validation, review and editing. All authors have read and agreed to the published version of the manuscript.

**Funding:** This work has been funded by the "Etorkizuna Eraikiz" program of the Gipuzkoa Provincial Council.

**Acknowledgments:** Especial thanks to Barys Shyrokau from TU Delft and to the research group Intelligent Vehicles for their support. Thanks to Manu Salazar for his help with the graphical abstract.

**Conflicts of Interest:** The authors declare no conflict of interest.

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
