# Peer review of "Comparison of Typical Controllers for Direct Yaw Moment Control Applied on an Electric Race Car"

_vehicles, doi:10.3390/vehicles3010008_

Round 1

Reviewer 1 Report

This manuscript compared the controllers on a modeled race car in terms of the robustness of the controllers, steering behavior, use of the friction circle, lap time on a track. The topic is very interesting, there are few concerns need to be further clarified.

1, The vehicle modeling review may be not sufficient, there are modeling approaches to accurately describe the vehicle multibody dynamics. It will be better if more related references can be reviewed.

2, The fuzzy controller and the optimal controller are also used for vehicles, related references may need to be included for comparison.

3, References needs to be cited for reader's conveniency if the models or the equations are provided in the manuscript.

Author Response

"1. The vehicle modelling review may be not sufficient" 

We thank the comment of the reviewer. In our opinion, despite there are better models for vehicles, since most of these controllers require to solve the model in real time, it is more reasonable to use the so-called bicycle model that is sufficiently complex to account for many of the characteristics of the vehicle but can be solved in real-time, as most stability control systems that can be found on passenger vehicles do.

In addition, we preferred to focus on the comparison of controllers instead of merging it with vehicle models. The different cases would have a difficult interpretation.

Just to clarify this assertion, the Carmaker model is a detailed vehicle model and the simplified bicycle model version is used only for the controllers.

"2. The fuzzy controller and the optimal controller are also used for vehicles, related references may need to be included for comparison."

There were several references to fuzzy controllers in Table 4. These references have also been included in the main text for convenience.

3. References needs to be cited for reader's convenience if the models or the equations are provided in the manuscript.

We have included the required reference in the manuscript.

Reviewer 2 Report

This paper compares different controller architectures for Torque Vectoring. In fact, a systematic comparison of the main control approaches (PID, LQR, SMC, MPC, and NMPC) and their performance is evaluated on two race tracks in this paper.

The paper is very well documented and written, and has original elements, and should be published. However, there are some aspects which have to be addressed.

  1. The paper should include a section of conclusions that reveals results with generic value.
  2. The text and English language in the paper have to checked, edited and corrected.

Author Response

1) "The paper should include a section of conclusions that reveals results with generic value".

We have included a conclusions section sating the most interesting results of the manuscript. 

2) "The text and English language in the paper have to checked, edited and corrected".

We have thoroughly revised the grammar and updated the manuscript accordingly.